# Autofluorescence Imaging Reflects the Nuclear Enlargement of Tumor Cells as well as the Cell Proliferation Ability and Aberrant Status of the *p53,* Ki-67, and *p16* Genes in Colon Neoplasms

**DOI:** 10.3390/molecules24061106

**Published:** 2019-03-20

**Authors:** Kentaro Moriichi, Mikihiro Fujiya, Yu Kobayashi, Yuki Murakami, Takuya Iwama, Takehito Kunogi, Takahiro Sasaki, Masami Ijiri, Keitaro Takahashi, Kazuyuki Tanaka, Aki Sakatani, Katsuyoshi Ando, Yoshiki Nomura, Nobuhiro Ueno, Shin Kashima, Katsuya Ikuta, Hiroki Tanabe, Yusuke Mizukami, Yusuke Saitoh, Toshikatsu Okumura

**Affiliations:** 1Division of Gastroenterology and Hematology/Oncology, Department of Medicine, Asahikawa Medical University, Asahikawa 078-8510, Japan; morimori@asahikawa-med.ac.jp (K.M.); kobayu@asahikawa-med.ac.jp (Y.K.); yuuki1228@asahikawa-med.ac.jp (Y.M.); ganmatakuya@asahikawa-med.ac.jp (T.I.); kunogi@asahikawa-med.ac.jp (T.K.); taka-sas@asahikawa-med.ac.jp (T.S.); m-ijiri@asahikawa-med.ac.jp (M.I.); ktakaha@asahikawa-med.ac.jp (K.T.); kazuyuki@asahikawa-med.ac.jp (K.T.); sakatani@asahikawa-med.ac.jp (A.S.); k-ando@asahikawa-med.ac.jp (K.A.); nomuzo@asahikawa-med.ac.jp (Y.N.); u-eno@asahikawa-med.ac.jp (N.U.); shin1014@asahikawa-med.ac.jp (S.K.); ikuta@asahikawa-med.ac.jp (K.I.); tant@asahikawa-med.ac.jp (H.T.); mizu@asahikawa-med.ac.jp (Y.M.); okumurat@asahikawa-med.ac.jp (T.O.); 2Digestive Disease Center, Asahikawa City Hospital, Asahikawa 070-8610, Japan; y_saito@city.asahikawa.hokkaido.jp

**Keywords:** autofluorescence imaging, colon neoplasm, N/C ratio, methylation, tumor cell proliferation

## Abstract

Background: Autofluorescence imaging (AFI) is useful for diagnosing colon neoplasms, but what affects the AFI intensity remains unclear. This study investigated the association between AFI and the histological characteristics, aberrant methylation status, and aberrant expression in colon neoplasms. Methods: Fifty-three patients with colorectal neoplasms who underwent AFI were enrolled. The AFI intensity (F index) was compared with the pathological findings and gene alterations. The F index was calculated using an image analysis software program. The pathological findings were assessed by the tumor crypt density, cell densities, and N/C ratio. The aberrant methylation of *p16*, *E-cadherin*, *Apc*, *Runx3*, and *hMLH1* genes was determined by a methylation-specific polymerase chain reaction. The aberrant expression of *p53* and Ki-67 was evaluated by immunohistochemical staining. Results: An increased N/C ratio, the aberrant expression of *p53*, Ki-67, and the altered methylation of *p16* went together with a lower F index. The other pathological findings and the methylation status showed no association with the F index. Conclusions: AFI reflects the nuclear enlargement of tumor cells, the cell proliferation ability, and the altered status of cell proliferation-related genes, indicating that AFI is a useful and practical method for predicting the dysplastic grade of tumor cells and cell proliferation.

## 1. Introduction

Colon cancer is a common tumor-causing disease and one of the most frequent causes of death in both Eastern and Western countries. With regard to molecular alterations, gastrointestinal cancers arise due to the accumulation of genetic alterations in epithelial cells during neoplastic transformation [1,2]. In this sequential process, epigenetic modifications, particularly DNA hypermethylation, in tumor-suppressor genes regulating cell proliferation, apoptosis, angiogenesis, and differentiation are commonly observed, in addition to genetic alterations [3,4]. These alterations and aberrant methylations of the tumor-suppressor genes, particularly cell cycle-associated genes, lead to an increase in cell densities and an enlargement of the tumor cell nucleus [5,6], which are all pivotal findings for histologically evaluating the dysplastic grade of gastrointestinal neoplasms [7]. Predicting the cell proliferation ability is useful for judging the malignant behaviors of tumors and determining the therapeutic strategy for individual patients with colorectal neoplasms.

Advanced endoscopic techniques have improved the diagnostic accuracy concerning the detection and differentiation of gastrointestinal neoplasms [8,9,10,11]. Autofluorescence imaging (AFI) is one of these novel technologies, which is efficient for detecting cancers in the esophagus, stomach, and colon [12,13,14], as well as dysplasia in Barrett’s esophagus [15,16,17] and ulcerative colitis [18,19,20]. Regarding the characterization of colon lesions, AFI was demonstrated to improve the diagnostic accuracy for differentiating colon neoplasms from hyperplastic polyps [21,22]. In addition, our recent study revealed that the fluorescence intensity of AFI is inversely proportional to the dysplastic grade of the colon adenoma [23]. As a result, AFI is considered to reflect the characteristic abnormalities caused by gastrointestinal neoplasms, although the effects of these histological findings and gene alterations on AFI findings are still poorly understood and insufficiently studied.

The present study investigates whether or not AFI scans are associated with the histological characteristics, the cell proliferation status, an altered expression of p53 protein, and the aberrant methylation of representative cancer-related genes, including *p16*, *E-cadherin*, *Apc*, *Runx3,* and *hMLH1,* in colon neoplasms.

## 2. Results

### 2.1. The Relationship between the F index and the Histological Characteristics of the Neoplasms

Six, 19, four, 10, and 14 lesions were histologically diagnosed as hyperplastic polyps, low and high grade adenomas, cancer in situ, and cancers with submucosal invasion, respectively. Although a univariate analysis showed that the F index showed no association with the tumor crypt and inflammatory cell density (Figure 1A,B), the tumor cell density and N/C ratio were significantly proportional to the F index (*p* < 0.05; Figure 1C,D). The multivariate analysis to identify associations between the F index and all pathological factors revealed that the N/C ratio was the only factor that independently affected the F index (Table 1). This suggests that AFI reflects the nuclear enlargement of tumor cells.

### 2.2. Relationships between the F index, Cell Proliferation Ability, and Aberrant Expression of p53 Proteins

To assess the cell proliferation ability, immunohistochemical staining with anti-Ki-67 antibodies was performed. The F index in the Ki-67 > 20 group was significantly lower than that in the Ki-67 ≤ 20 group (Figure 2A). The expression of p53 proteins, which are known to be gatekeeper genes in the cell cycle and are aberrantly expressed when the gene is mutated in tumors, was subsequently assessed using immunohistochemistry. The F index in the cells with aberrant expression of p53 proteins was significantly lower than that in the cells with normal expression (Figure 2B). This suggests that the F index predicts the capacity for cell proliferation in colorectal cancer cells.

### 2.3. Relationship between the F index and the Aberrant Methylation of Tumor-Related Genes

Among the 53 lesions, the aberrant methylation of *p16*, *Apc*, *hMLH1*, *E-cadherin,* and *Runx3* was detected in 7/50 (14.0%), 12/51 (23.5%), 1/39 (2.6%), 9/50 (18.0%), and 15/29 (51.7%) lesions, respectively. The F index of the lesions which exhibited the aberrant methylation of *p16* (0.53 ± 0.12) was significantly lower than that of neoplasms without the aberrant methylation of *p16* (0.71 ± 0.21) (Figure 3A). Conversely, there was no significant difference in the F index in the lesions with and without the aberrant methylation of *Apc, hMLH1, E-cadherin,* and *Runx3* (Figure 3B–E) (Figure 4, case presentation).

## 3. Discussion

The present study was the first to demonstrate that the fluorescence intensity of AFI reflects the nuclear enlargement of tumor cells, showing that AFI can be used to predict the dysplastic grade of colorectal tumors simply by assessing the strength of the magenta color. It is noteworthy that the intensity of fluorescence captured by AFI affects the cellular atypia as well as changes in the *p53* and *p16* genes, which are essential for regulating the cell cycle, and are suppressors of the initiation and promotion of cancer [24,25,26]. While AFI could not directly detect the genes themselves, this procedure still predicted the aberrant status of the *p53* and *p16* genes by measuring the attenuation of the fluorescence intensity. Therefore, although the normal area of the colon appeared green, which corresponded to an F index of approximately 1.2 or more, colon neoplasms detected as a strong intensity of the magenta color by AFI potentially possess alterations in cancer-related genes and should be aggressively treated by endoscopic resection or surgery.

It is well known that chromatin is increased in the nuclei of colorectal tumor cells and that the changes are proportional to the dysplastic grade of the tumor. This study showed that an increased N/C ratio significantly diminished the fluorescence intensity captured by AFI. The cell nucleus is composed of nucleic acids and proteins such as histones and transcription factors, whereas the cytosol contains a much lower ratio of these molecules compared to the mucus. These two major components of cells are thought to be responsible for the different permeability to fluorescence by the intestinal tissues, and thus a high N/C ratio diminishes the fluorescence intensity captured by AFI due to the high densities of nucleic acids and proteins. A high N/C ratio in tumor cells is a pivotal histological finding indicating a high rate of cell proliferation. Consequently, AFI is thought to be a feasible tool to determine the malignant potential of colon neoplasms.

While the fluorescence intensity captured by AFI was decreased in the lesions with an aberrant status of the *p53* and *p16* genes, which are essential for regulating the cell cycle and the proliferation of cancer [24,25,26], the aberrant methylation of *Apc, E-cadherin*, *Runx3,* and *hMLH1* did not influence the fluorescence intensity of AFI in colon neoplasms. Generally, *Apc* plays an important role in the adenoma-carcinoma sequence. *Apc* is widely recognized as a tumor-suppressive gene and an alteration of this gene causes tumor growth by affecting the Wnt and other signaling pathways [27,28,29]. Although the aberrant methylation status of *Apc* did not affect the F index, the other alterations, including the mutation, may affect the F index. *E-cadherin* is an adhesion molecule expressed in the tight junctions between cells and is frequently detected in colon cancer cells [30,31,32]. A decreased expression of *E-cadherin* changes the structure and density of tumor crypts. *Runx3* is a transcription factor related to the transforming growth factor-β (TGF-β) signaling pathway and plays important roles in mammalian development [33]. The aberrant methylation of *Runx3* is frequently detected in colon cancer cells [34,35], resulting in structural alterations and a decreased density of tumor crypts. Because the aberrant methylation of *E-cadherin* and *Runx3* appears to affect the crypt density while only slightly affecting the N/C ratio of tumor cells, changes in these genes showed no correlations with the fluorescence intensity of the AFI images in the present study.

Croce summarized the endogenous fluorophores that are repeatedly exploited as intrinsic biomarkers in autofluorescence studies, including aromatic amino acids, cytokeratins, collagen/elastin, NAD(P)H, flavins, fatty acids, vitamin A, protoporpyrin IX, and lipofuscins [36]. Among these, considering the wavelength of excitation and emission, collagen/elastin, flavins, and lipofuscins can be detected by endoscopic AFI.

The collagens in the submucosal layer are thought to be more responsible for the autofluorescence from the human colorectal wall than other fluorophores, because it is known that collagens are enriched in the colorectal submucosal layer [37], and the AFI system used in our study was adjusted to the autofluorescence emitted from submucosal collagen. On the other hand, flavins are coenzymes in redox reactions and are thought to be correlated with alterations of energy metabolism, inflammation, and carcinogenesis. The various types of lipofuscins, including proteins, lipids, and retinoids, are thought to be correlated with alterations of the degree oxidation and cell stemness. These molecules are considered to be associated with abnormalities of *p16*, *p53,* and Ki-67, which are closely associated with tumorigenesis. Thus, flavins and lipofuscins might be present in tumor cells, affecting the autofluorescence intensity of the tumor lesion. However, no study has reported the enrichment of these two molecules in the colorectal wall, suggesting that these molecules have relatively little influence on the autofluorescence intensity.

In conclusion, the present study showed that AFI reflects the nuclear enlargement of tumor cells, which is a key finding for assessing the histological dysplastic grade and the Ki-67 expression of tumor cells, which reflects the cell proliferation ability. Furthermore, the intensity of fluorescence captured by AFI is associated with the aberrant status of cell proliferation-related genes, including *p53* and *p16*. Our results suggest that AFI can predict the dysplastic grade of tumor cells, as well as some abnormalities in the genes related to cell proliferation. Accordingly, colon neoplasms, depicted as a strong magenta area by AFI, potentially possess some malignant potential, both histologic and genetic, and are considered to be indicated for endoscopic or surgical resection. The quantification of the fluorescence intensity in a real-time fashion can immediately provide objective information for determining the indications for performing a resection of the colon neoplasm.

## 4. Materials and Methods

### 4.1. Samples

This study has been registered with the University Hospital Medical Information Network (UMIN000002019). Written informed consent was obtained from all patients enrolled, and the study was approved by the institutional review board of Asahikawa Medical University. Fifty-three patients with colorectal neoplasms as diagnosed at Asahikawa Medical University Hospital, who underwent an AFI examination (CF-FH260AZI, Olympus medical systems, Tokyo, Japan), were enrolled in this study.

### 4.2. AFI

A high-definition colonoscope (CF-FH260AZI; Olympus Corporation) containing two Charge Coupled Devices (CCDs, one for high resolution endoscopy/narrow band imaging and one for AFI) as well as an Olympus Lucera Spectrum video processor and a high-definition monitor were used. White light is emitted from a 300-W xenon lamp as the light source and then separated with a rotary filter. AFI uses blue light (wavelength 390–470 nm) for excitation and green light (wavelength 540–560 nm) for reflection. A barrier filter allows the passage of light to the CCDs with wavelengths between 500 nm and 630 nm only, constituting autofluorescence emission and green reflectance. A pseudocolor image is produced by allocating the detected and amplified autofluorescence signal to the green (G) channel and the reflected signal of green light to the red (R) and blue (B) channels at a ratio of 1 to 0.5 [11,38].

The strength of fluorescence emitted from the intestinal tissues in the 53 AFI images were quantified with an image analysis software program (provided by Olympus Medical Systems (not commercially available)). Briefly, we detected the region of interest using white light endoscopy and subsequently observed and took pictures of the regions with AFI. The AFI images were converted into bitmap-formatted pictures, and the tumor area on the pictures was manually traced as a rectangular shape using the software program (Figure 5). The signal density of either red or green in the traced area was measured and converted to a reverse gamma value, which corresponded to the strength of the output from each CCD signal. This software program removes gamma compensation, which enabled us to analyze the real color of the lesion.

The main source of autofluorescence is known to be collagen and elastin in the submucosal layer. When the mucosal layer is thickened, both the excitation light and autofluorescence are diminished during the permeation of the mucosal layer, leading to the weakness of the autofluorescence captured by CCD. The thickness of the mucosal layer is dependent on the extension status of the intestinal lumen. If AFI is obtained with inappropriate extension of the lumen due to insufficient filling of air, autofluorescence is weakened, even in normal areas. We therefore developed the following procedure to quantify green autofluorescence intensity with respect to the red reflectance [39] and applied the procedure in this study.

The ratio of the reverse gamma value of green (fluorescence) divided by that of red (reflex) was defined as the fluorescence index (F index) [39,40] (F index = green (fluorescence)/red (reflection)). This quantification was performed by an endoscopist who was not aware of any of the patients’ endoscopic or histological information. The F index was compared with the histological characteristics and the methylation status of genes in all the colon neoplasms examined.

### 4.3. Histological Assessment

Histological specimens were obtained from the colon lesions by target biopsy, endoscopic resection, or surgery, immediately fixed with 10% formalin for 24 h, and then embedded in paraffin 24 h after fixation. Four micrometer sections were prepared and stained with hematoxylin and eosin. The dysplastic grades were assessed according to the Vienna classification [7] by one pathologist who was blinded to the clinical information of the patients.

The tumor crypt density was estimated in a representative section of the lesions by measuring the area of the tumor crypt in one field of a 200× image. Similarly, the tumor cells were counted in one field of a 200× image. The area of the nucleus was measured and divided by the area of the cytoplasm to obtain the N/C ratio. 

### 4.4. Immunohistochemistry

Immunohistochemistry was performed to examine the status of cell proliferation and the expression of p53 protein using anti-Ki-67 (MIB-1; DAKO, Glostrup, Denmark) and anti-p53 mouse monoclonal antibodies (DO-7; DAKO) as primary antibodies, respectively. Following deparaffinization and rehydration, the endogenous peroxidase activity was blocked with 0.6% H_2_O_2_ in methanol for 25 min. The slides were then treated with the antigen-retrieval technique using microwave oven heating in 10 mM citrate buffer (pH 6.0) for 20 min. The container was allowed to cool at room temperature for 20 min. After blocking any nonspecific reactions with 10% goat serum in phosphate buffered saline (PBS), the sections were incubated with the primary antibodies at 4 °C overnight. This step was followed by sequential incubation with biotin-labeled goat anti-rabbit IgG and avidin biotin complex reagents (Vector Laboratories, Burlingame, CA, USA). The biotinylated secondary antibodies were diluted at 1:100 for 30 min at room temperature. The sections were visualized with diaminobenzidine-H_2_O_2_ solution and counterstained with hematoxylin. The quantification of Ki-67 has already been described and a description of the meaning of Ki-67 has been provided [41]. According to Agdo’s report [42], when immunochemical staining using an MIB-1 antibody showed that >20% of the cells in three randomly selected fields were Ki-67-positive, the lesion was classified into the Ki-67 > 20 group. Conversely, when ≤20% of cells in three randomly selected fields were Ki-67-positive, the lesion was classified into the Ki-67 ≤ 20 group. The p53 expression was considered positive when >10% of the tumor cells exhibited specific staining [42].

### 4.5. DNA Preparation and Methylation-Specific PCR

Ten micrometer sections were prepared for DNA extraction. The sample was precisely micro-dissected under microscopic visualization, using a P.A.L.M MG III Laser Capture Microdissection System (MEIWAFOSIS, Osaka, Japan) to avoid DNA contamination by inflammatory or stromal cell nuclei [43,44,45]. The methylation status of each gene in the samples was then analyzed by a methylation-specific PCR (MS-PCR) [46,47]. Briefly, purified DNA samples were chemically modified by sodium bisulfite with the CpGenome^TM^ DNA Modification Kit (Chemicon International, Temecula, CA, USA) to convert all unmethylated cytosines to uracils, while leaving the methylcytosines unaltered. The bisulfite-modified DNA was amplified using primer pairs that specifically amplify either the methylated or unmethylated sequences of the five target genes, *p16*, *Apc*, *hMLH1*, *E-cadherin,* and *Runx3*. MS-PCR was performed in duplicate using the primer sequences for the methylated and unmethylated forms of all genes, and the annealing temperatures are summarized in Table 2 [46,48,49,50]. The PCR was conducted in 25 µL reaction volumes containing 1XPCR buffer, 2 mmol/L MgCl2, 0.25 mmol/L each of the dNTPs, 5 pmol of the primers, and 1 unit of AmpliTaq Gold polymerase (Perkin Elmer, Waltham, MA, USA). The PCR conditions were 10 min at 95 °C, followed by 40 cycles of denaturing at 95 °C for 45 s, annealing for 45 s, and then extension at 72 °C for 1 min. CpGenome Universal Methylated DNA (Intergen, Purchase, NY, USA) and reagent blanks were used as positive and negative controls, respectively, in each experiment. Amplified DNA products were analyzed by a HAD-GT12 multicapillary electrophoresis system (eGene, Irvine, CA, USA) using a 12-capillary gel-cartridge (GCK5000). This system is an automated DNA fragment analyzer offering high resolution [51].

### 4.6. Statistical Analyses

The Mann−Whitney U-test and χ^2^ test were applied for the statistical analyses of the relationship between the F index and histological findings and the status of aberrant gene methylation. To estimate the independent factors of the histological findings affecting the F index, a multiple regression analysis was applied as a multivariate analysis. A value of *p* < 0.05 was considered to be statistically significant.

## Figures and Tables

**Figure 1 molecules-24-01106-f001:**
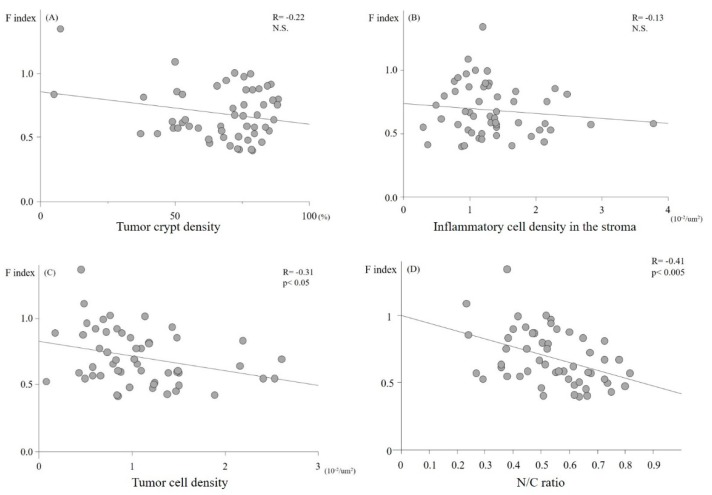
The relationship between the autofluorescence imaging (AFI) fluorescence intensity (F index) and the histological characteristics of the tumor. No significant correlation was detected between the tumor crypt density (**A**) or the inflammatory cell density in the stromal (**B**) of the lesions and the F index. In contrast, the tumor cell density and the N/C ratio showed significantly inverse correlations with the F index (**C**,**D**).

**Figure 2 molecules-24-01106-f002:**
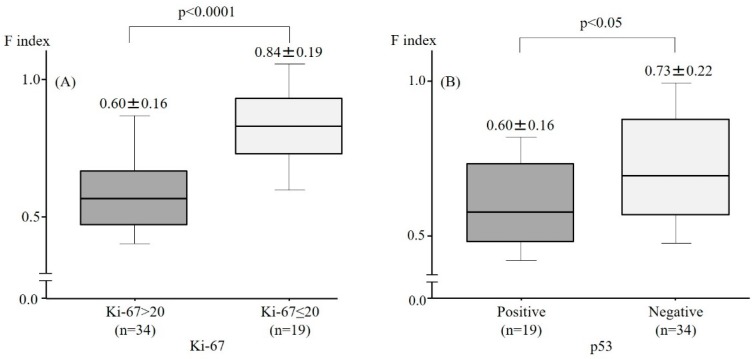
The relationship between the F index, cell proliferation ability, and aberrant expression of p53. The F index in the Ki-67 > 20 group and the aberrant expression of p53 proteins group was significantly lower than that observed in the Ki-67 ≤ 20 group (**A**,**B**).

**Figure 3 molecules-24-01106-f003:**
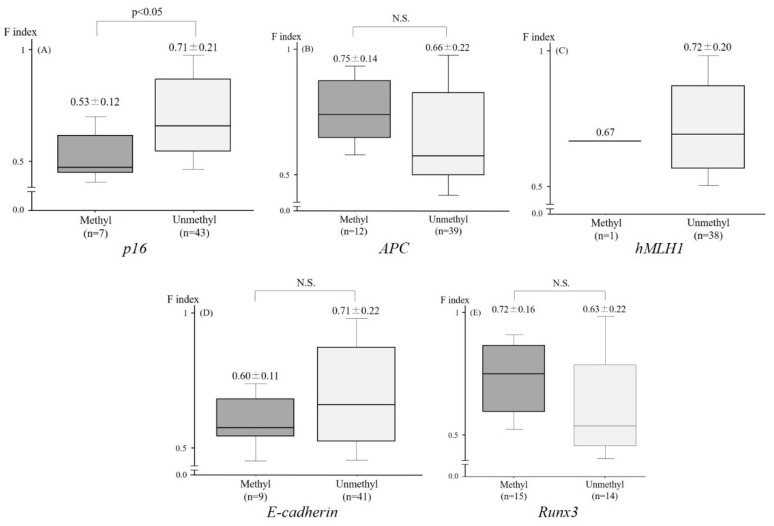
The relationship between the F index and the aberrant methylation of tumor-related genes. The F index of the lesions which exhibited the aberrant methylation of *p16* was significantly higher than that of tumors without the aberrant methylation of *p16* (**A**). Conversely, there was no significant difference in the F index between lesions with and without the aberrant methylation of *Apc* (**B**), *hMLH1* (**C**), *E-cadherin* (**D**), and *Runx3* (**E**).

**Figure 4 molecules-24-01106-f004:**
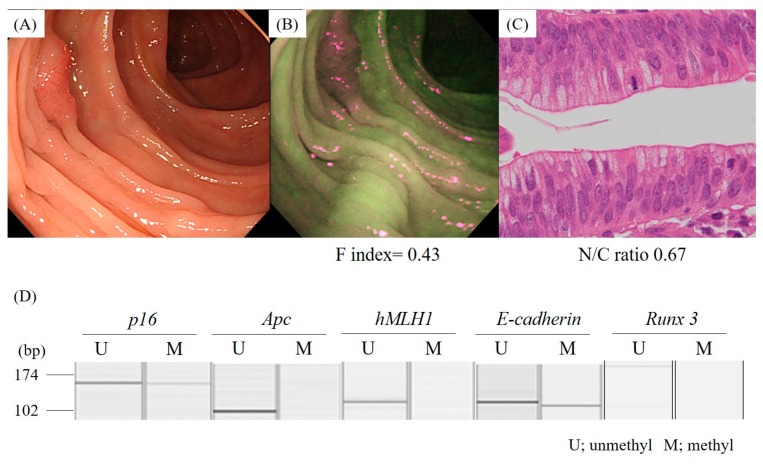
A representative case of colon cancer with the aberrant methylation of the *p16* and *E-cadherin* genes. A conventional colonoscopy revealed an elevated lesion (**A**). AFI detected the lesion as a strong magenta area (**B**). A high N/C ratio (0.67) was observed on the histological specimen of the lesion, which was diagnosed to be a cancer in situ (**C**). MS-PCR showed the aberrant methylation of the *p16* and *E-cadherin* gene promoters (**D**).

**Figure 5 molecules-24-01106-f005:**
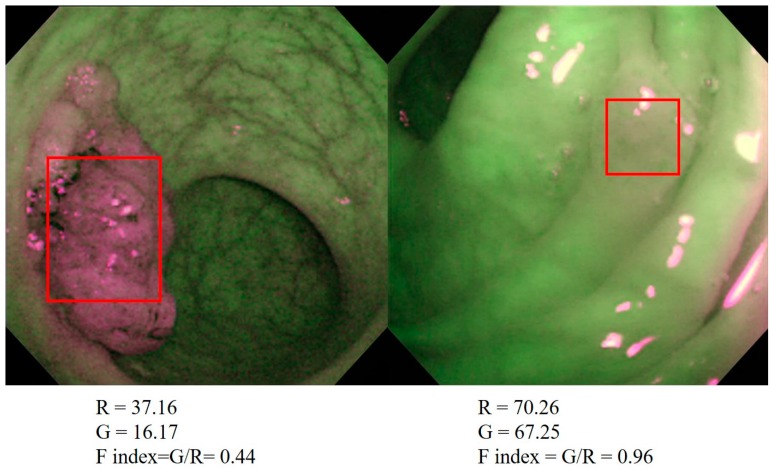
Quantification of AFI images with a software program used to calculate the F index. AFI images were converted into bitmap-formatted pictures and the tumor area was manually traced. The signal density of either red or green in the traced area was measured and converted to a reverse gamma value, which corresponded to the strength of the output from each CCD signal. The ratio of the reverse gamma value of the green (fluorescence) divided by that of the red (reflection) was defined as the F index.

**Table 1 molecules-24-01106-t001:** The multivariate analysis between the F index and pathological findings extracted by univariate analysis.

Variables	Regression Coefficient	95% CI
Tumor crypt density	−0.072	−0.346, 0.202
Inflammation cell density	1.490	−7.798, 0.778
Tumor cell density	−5.330	−17.371, 6.710
N/C ratio	−0.522	−0.960, −0.085

CI: Confidence Interval.

**Table 2 molecules-24-01106-t002:** For methylation-specific PCR.

Primer Name	Primer Sequence (5′–3′)	Product Size (bp)	Annealing Temperature (°C)
*hMLH1*-U (F)	TTTTGATGTAGATGTTTTATTAGGGTTGT	124	60
*hMLH1*-U (R)	ACCACCTCATCATAACTACCCACA		
*hMLH1*-M (F)	TATATCGTTCGTAGTATTCGTGT	153	60
*hMLH1*-M (R)	TCCGACCCGAATAAACCCAA		
*E-cadherin*-U (F)	TGGTTGTAGTTATGTATTTATTTTTAGTGGTGTT	120	60
*E-cadherin*-U (R)	ACACCAAATACAATCAAATCAAACCAAA		
*E-cadherin*-M (F)	TGTAGTTACGTATTTATTTTTAGTGGCGTC	112	64
*E-cadherin*-M (R)	CGAATACGATCGAATCGAACCG		
*p16*-U (F)	TTATTAGAGGGTGGGGTGGATTGT	151	60
*p16*-U (R)	CAACCCCAAACCACAACCATAA		
*p16*-M (F)	TTATTAGAGGGTGGGGCGGATCGC	150	65
*p16*-M (R)	GACCCCGAACCGCGACCGTAA		
*APC*-U (F)	GTGTTTTATTGTGGAGTGTGGGTT	108	61
*APC*-U (R)	CCAATCAACAAACTCCCAACAA		
*APC*-M (F)	TATTGCGGAGTGCGGGTC	98	63
*APC*-M (R)	TCGACGAACTCCCGACGA		
*Runx3*-U (F)	TTATGAGGGGTGGTTGTATGTGGG	221	56
*Runx3*-U (R)	AAAACAACCAACACAAACACCTC		
*Runx3*-M (F)	TTACGAGGGGCGGTCGTACGCGGG	221	66
*Runx3*-M (R)	AAAACGACCGACGCGAACGCCTCC		

U; unmethylated sequence, M; methylated sequence, APC; adenomatous polyposis coli, F; forward, R; reverse.

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
