# Peer review of "Autofluorescence Imaging Reflects the Nuclear Enlargement of Tumor Cells as well as the Cell Proliferation Ability and Aberrant Status of the p53, Ki-67, and p16 Genes in Colon Neoplasms"

_molecules, 2019, doi:10.3390/molecules24061106_

Round 1

Reviewer 1 Report

In this manuscript the authors use auto-fluorescence imaging (AFI) of colon tissue via a medical colonovideoscope to identify neoplasms. The AFI uses false color representation to encode fluorescence intensity where a magenta color represents low auto-fluorescence vs. green colors indicate (normal) higher fluorescence intensity. For analysis of this color change the ratio of the green to the red color value of each pixel are is calculated (the so called F-index). These values were averaged over regions of interest that are indicative for possible tumor development.

At first, the determined F-index was analyzed for correlations to specific histological parameters of colon neoplasms, like tumor crypt density, inflammatory cell density, tumor cell density, and the ratio of nuclear to cytosolic area (N/C ratio). Furthermore, a classification with respect to proliferation ability indicators Ki-67 and p53 was done as well as correlation to DNA-methylation of five specific genes.

The conclusion of the study is that the F-index is indicative of an increased N/C ratio, Ki-67 expression as well as p53 expression and methylation of the p16 gene.

The manuscript describes quite well the methods and procedures that were used in this study. Similar studies have been done before. Here, the authors try to find a link between the reduced auto-fluorescence and histological and biochemical findings.

The numbers of cases for some of the tests are quite low so the results will have to be corroborated by a study with more patients. However the findings are promising. However, the more care should be taken when presenting the conclusions of the study. The present statements suggests, that the F-index indicates all of the four statistical trends. Instead, the conclusion should rather be like: An increased N/C ratio, KI-67 expression, p53 expression, and methylation of p16 go together with a lower F-index. Since there was no test for a mutual dependence of these factors, a lower F-index can have any the backgrounds and potentially others. I suggest to revise the statements in the Abstract and the conclusions sections accordingly. It would be quite helpful if the authors could include some statements of the distribution of the F-index in healthy tissue.

Reviewer 2 Report

The authors present work correlating the intensity of autofluorescence to known genetic and morphologic markers of colon cancer. Autofluorescence imaging is an emerging technology that offers exciting avenues for improved cancer detection. This manuscript offers clinically relevant results from a large cohort of patients (53). However, there are significant revisions necessary prior to publication. 

The authors reference the "F-index" to quantify "green" autofluorescent intensity with respect to "red" reflectance. However, the manuscript provides no motivation for asserting the F-index over just the intensity or reflectance alone (or another fluorescent metric). 

There are few specifics on the imaging parameters of the endoscope. Which wavelengths were used for excitation and emission? Why those wavelengths? Which CCD camera was used? Were the images corrected for background noise?

There is no mention of how the images were analyzed. Which software was used for image acquisition and analysis? How were the regions of interest selected?

Fig. 1 shows separate plots of the F index versus morphologic features. However, these plots are less convincing without including a multivariate analysis incorporating all parameters to confirm that the parameter that offers the best separation between tumor and non-tumor is the F index.

Table 1 should include all variables (not just the significant variables).

In Fig. 2, did Ki-67-expressing cells in the "Negative" group feature no proliferating cells? How was Ki-67 quantified?  

It's not clear how the F-index can be compared to morphologic or genetic parameters. Given the heterogeneous and non-specific nature of autofluorescence, what conclusions can be drawn from its use without first understanding the molecular nature of the fluorescence?

There are previous studies incorporating morphologic features and autofluorescence, thus it's unclear how this manuscript improves on existing work.

Round 2

Reviewer 2 Report

The authors present work correlating the intensity of autofluorescence to known genetic and morphologic markers of colon cancer. Autofluorescence imaging is an emerging technology that offers exciting avenues for improved cancer detection. This manuscript presents potentially new results in detecting colon cancer and correlating the fluorescence measurements to immunohistochemistry and morphology. However, there are significant revisions necessary prior to publication. 

Several imaging parameters should be included: endoscope manufacturer and model, CCD camera manufacturer and model, image analysis software name, exposure time for each image acquired by the camera, and if any camera gain was used - the amount of gain should also be noted.

What is the difference in signal between background and autofluorescence intensity? 

The manuscript originally notes that regions of interest were selected by manual traced as a rectangular shape. Given the amorphous nature of tumors, why wasn't a free-form tracing method used to more accurately trace the tumor margins?

It's unclear what is meant by "reverse gamma value." Is this still a measure of overall photons detected by the CCD camera? 

Fig. 1 shows separate plots of the F index versus morphologic features. What is the R value for tumor cell density versus N/C ratio? Does it exceed that acquired with the F-index? This would be a helpful plot to understand the significance or motivation for using the F-index.

It's not clear how the F-index can be correlated to genetic parameters. The manuscript notes, "While AFI could not directly detect the genes themselves, this procedure still predicted the aberrant status of the p53 and p16 genes by measuring the attenuation of the fluorescence intensity." However, no follow-up controls were performed to determine the nature of this correlation. For instance, if p53 or p16 were modified differently, does the F-index change accordingly?

Fig. 2 presents "Positive" and "Negative" groups of Ki-67 expression, but these groups should conform to the labels used throughout the paper (Ki-67 >20 group and Ki-67<20 group, respectively).

There are previous studies incorporating morphologic features and autofluorescence, (Skala, et al. 2007 specifically measured the in vivo optical redox ratio and nuclear to cytoplasmic ratio of cancerous and precancerous lesions), and it's not clear how this paper advances the field without further control studies. 
